# Viral Diversity in Benthic Abyssal Ecosystems: Ecological and Methodological Considerations

**DOI:** 10.3390/v15122282

**Published:** 2023-11-21

**Authors:** Umberto Rosani, Cinzia Corinaldesi, Gabriella Luongo, Marco Sollitto, Simeone Dal Monego, Danilo Licastro, Lucia Bongiorni, Paola Venier, Alberto Pallavicini, Antonio Dell’Anno

**Affiliations:** 1Department of Biology, University of Padova, Via U. Bassi 58/b, 35121 Padova, Italy; paola.venier@unipd.it; 2Department of Materials, Environmental Sciences and Urban Planning, Polytechnic University of Marche, Via Brecce Bianche, 60131 Ancona, Italy; c.corinaldesi@staff.univpm.it; 3Department of Life and Environmental Sciences, Polytechnic University of Marche, Via Brecce Bianche, 60131 Ancona, Italy; g.luongo@pm.univpm.it; 4Department of Life Sciences, University of Trieste, Via Licio Giorgeri 5, 34127 Trieste, Italy; marco.sollitto@famnit.upr.si (M.S.); pallavic@units.it (A.P.); 5Faculty of Mathematics, Natural Sciences and Information Technologies, University of Primorska, 6000 Koper, Slovenia; 6Laboratorio di Genomica ed Epigenomica, AREA Scienze Park, Padriciano 99, 34149 Trieste, Italy; simeone.dalmonego@areasciencepark.it (S.D.M.); danilo.licastro@areasciencepark.it (D.L.); 7Consiglio Nazionale delle Ricerche, Istituto di Scienze Marine, Tesa 104–Arsenale, Castello 2737/F, 30122 Venezia, Italy; lucia.bongiorni@ve.ismar.cnr.it

**Keywords:** deep-sea viromes, shotgun viromics, ssDNA viruses

## Abstract

Viruses are the most abundant ‘biological entities’ in the world’s oceans. However, technical and methodological constraints limit our understanding of their diversity, particularly in benthic abyssal ecosystems (>4000 m depth). To verify advantages and limitations of analyzing virome DNA subjected either to random amplification or unamplified, we applied shotgun sequencing-by-synthesis to two sample pairs obtained from benthic abyssal sites located in the North-eastern Atlantic Ocean at ca. 4700 m depth. One amplified DNA sample was also subjected to single-molecule long-read sequencing for comparative purposes. Overall, we identified 24,828 viral Operational Taxonomic Units (vOTUs), belonging to 22 viral families. Viral reads were more abundant in the amplified DNA samples (38.5–49.9%) compared to the unamplified ones (4.4–5.8%), with the latter showing a greater viral diversity and 11–16% of dsDNA viruses almost undetectable in the amplified samples. From a procedural point of view, the viromes obtained by direct sequencing (without amplification step) provided a broader overview of both ss and dsDNA viral diversity. Nevertheless, our results suggest that the contextual use of random amplification of the same sample and long-read technology can improve the assessment of viral assemblages by reducing off-target reads.

## 1. Introduction

Increasing evidence based on high-throughput sequencing (HTS) techniques indicates that marine viruses are highly diverse [1,2,3,4,5,6], and that benthic deep-sea environment can be a hotspot of viral diversity [7,8,9,10,11,12,13,14]. However, technical and methodological constraints still limit our understanding of the diversity of viruses present in deep-sea sediments. In fact, unlike cellular organisms which universally have double-stranded (ds) DNA genomes, viral genomes consist of either RNA or DNA molecules in ds or single-stranded (ss) forms, circular or linear, and encoded in single-unsegmented or multiple-segmented molecules, and display a remarkable diversity in terms of size and gene composition [15,16], hampering an unbiased overview of the viral diversity. So far, two different approaches have been used to analyse the diversity of DNA viruses in deep-sea sediments: one based on the extraction of the viral particles from the sediment and subsequent concentration of the viral-size fraction through filtration and/or ultracentrifugation, before DNA extraction and virome analysis [2,7,8,9] and the other based on the extraction from the sediment of total DNA to be used directly for high-throughput shotgun sequencing, without any separation step of viral particles [10,11,13,14].

Both these approaches have intrinsic limitations and can introduce biases in assessing the actual viral diversity. Indeed, viruses can be extracted from the sediment with a different efficiency [8,17,18] and the total amount of DNA recovered is often very low and requires a random amplification step before the library preparation and sequencing, which can result in an over-representation of ssDNA viruses [8]. At the same time, the extraction of total DNA from environmental samples can underestimate viral diversity [19], as sample processing and sequencing can limit the identification of a large fraction of viruses with a single-strand genome [10,11,12,13], which have been repeatedly reported as an important fraction of the overall viral diversity in deep-sea sediments [2,7,8,9]. At present, methodological advancements allow preparing libraries starting from very low amounts of DNA (e.g., from 1 pg to 1 ng of total DNA extracted from surface water and deep-sea sediments [20]), possibly circumventing the problem of the pre-amplification step. Moreover, besides short-read sequencing platforms, such as Illumina, long-read sequencing platforms can also be used to facilitate the reconstruction of viral genomes [21,22], despite being more error-prone [23].

Here, to evaluate advantages and limitations of analysing unamplified (native) and randomly amplified viral-enriched DNA samples, we applied short-read Illumina shotgun sequencing to two sample pairs obtained from the upmost sediment of two abyssal sites (ca. 4700 m depth) of the North-eastern Atlantic Ocean. Long-read Nanopore sequencing was applied to one amplified sample for comparison and the taxonomic composition of the DNA viromes obtained with the different methodologies was compared and discussed to expand our knowledge of viral diversity in the largest ecosystem of Earth.

## 2. Materials and Methods

### 2.1. Sediment Collection and Viral DNA Extraction

Undisturbed sediment samples were collected using a multi-corer between May and June 2018 at two sites (hereafter defined as Site 39, 48°58.939′ N and 16°33.026′ W, and Site 57, 48°58.964′ N and 16°32.866′ W) ca. 1.8 km from each other and at a depth of ca. 4700 m in the Abyssal Hill province, located 480 km southwest of Ireland in the North-eastern Atlantic Ocean. The Abyssal Hill province is characterized by an elevation of ~300 m from the abyssal plain (the Porcupine Abyssal Plain) and current speeds up to ~10 cm s^−1^ [24]. Sediments of the Abyssal Hill province are generally characterised by a coarser grain size compared to abyssal sediments [24,25] and the C input to the seafloor is estimated to be ca. 1.1 mmol C m^−2^ d^−1^, largely composed of labile organic matter [26]. The C inputs to the Abyssal Hill seafloor sustain high biomass and diversified megafauna assemblages (ca. 9 g m^−2^, largely represented by suspension feeders [27]), and a relatively high C consumption rate (0.479 mmol C m^−2^ d^−1^ vs. 0.340 mmol C m^−2^ d^−1^ of bacteria [26]). Sediment samples were collected on the flank of the abyssal hill at ca. 30 m depth above the abyssal plain with a median flank slope angle of 7.6° [25]. After recovery, the superficial sediment layer (0–1 cm) was snap-frozen and maintained at −80 °C until further use. Viruses were extracted from the sediment according to a previously published procedure [8]. Before virus extraction, a similar amount of the top 0–1 cm of sediment collected by three independent multi-corer deployments at each benthic site was pooled together to provide a more realistic view of the actual viral diversity present therein. Briefly, a total amount of 50 g of wet sediment was mixed with 50 mL of virus-free seawater (pre-filtered onto 0.02-μm-pore-size filters and autoclaved) into a sterile jar and aliquots of 2 mL of the sediment slurry were split into 50 sterile tubes to increase the viral recovery efficiency. Epifluorescence microscopy observations using Sybr Gold as stain revealed the lack of contamination by viral particles of seawater used to prepare the sediment slurries. Each sample was then supplemented with tetrasodium pyrophosphate solution (5 mM final concentration), incubated for 15 min in the dark and sonicated 3 times (Branson Sonifier 2200, 60 W, Thermofisher, Waltham, MA, USA) for 1 min. After centrifugation (800× *g* for 10 min), the supernatants were collected and the sediments were resuspended again with 4 mL of virus-free seawater, centrifuged (800× *g* for 10 min) and the supernatant collected. This step was repeated three times. The supernatants were combined and incubated with DNase I (1 U/mL, Merck, St. Luis, MO, USA) in the dark at room temperature for 15 min to remove extracellular DNA. The supernatant was pre-filtered through 0.2 μm pore size filters (Whatman Anopore, Merck, Darmstadt, Germany) to remove cellular organisms before concentrating viral particles onto 0.02 μm pore size Al_2_O_3_ filters (Anodisc; diameter 47 mm, Merck). Filters containing the collected viruses were subjected to DNA extraction by using the DNeasy PowerSoil Kit (Qiagen, Hilden, Germany) following manufacturer instructions. The viral DNA obtained was split into two aliquots, one directly used for library preparation and sequencing, whereas the other was subjected to amplification using the highly processive, proof-reading proficient, Phi29 polymerase (GenomiPhi kit, Merck) and random hexamer primers (Thermofisher), before further processing. Samples amplified with the GenomiPhi kit were then processed with the DNeasy PowerClean Cleanup Kit (Qiagen) following manufacturer instructions. The DNA concentrations in all samples were quantified fluorometrically with SYBR Gold on a NanoDrop ND-3000 (Thermofisher). The size distribution of the DNA samples was determined by microfluidics-based electrophoresis using a DNA high-sensitivity (HS) kit in an Agilent 2100 instrument (Agilent, Santa Clara, CA, USA).

### 2.2. Library Preparation and Illumina Short-Read Sequencing

The DNA used to prepare libraries for short read sequencing was fragmented with a BioruptorTM UCD-200 instrument (Diagenode, Denville, NJ, USA), using the tuning intensity and fragmentation times described below. Amplified DNA samples were fragmented for 15 min, with 45″/15″ on/off steps, at maximum intensity with samples maintained on ice. Unamplified DNA samples were fragmented with the same cycle for 2 min. DNA libraries were prepared using the Swift Accel-NGS 1S Plus DNA Library Kit (Swift BioSciences, Washtenaw County, MI, USA), following manufacturer recommendations. A starting DNA amount of 10 ng or approximately 20–50 pg was used for the amplified and unamplified samples, respectively (Table 1). The number of PCR cycles was set to 5 for amplified samples and 16 for the unamplified ones. DNA libraries were quantified with a Qubit 2.0 instrument (Thermofisher) using a DNA HS kit and the size distributions were determined as reported above. Libraries were pooled in equimolar quantities and sequenced on an Illumina NovaSeq6000 instrument (Area Science Park, Trieste, Italy), with a 150 bp paired-end layout.

### 2.3. Nanopore Library Preparation and Sequencing

The amplified DNA obtained from Site 57 has also been sequenced by means of long-read Oxford Nanopore Technology platform. It was not possible to apply the same technology to the non-amplified DNA since the minimal amount of required DNA is currently 1 µg. The amplified DNA used for long-read sequencing was not fragmented and the sequencing library was prepared with the standard ligation sequencing kit LSK109 (Oxford Nanopore Technologies, Oxford, UK), adjusting the DNA repair and end-prep incubation step to 20 min. Sequencing was conducted on a MinION equipped with a FLO-MIN106 (R 9.4.1) flowcell (Oxford Nanopore Technologies) for 48 h with the base calling mode set as active.

### 2.4. Sequencing Data Processing

Raw Illumina readouts were demultiplexed and reads were trimmed for quality and to remove sequencing adaptors using TrimGalore (https://github.com/FelixKrueger/TrimGalore (accessed on 1 May 2023)). The minimal quality was set to PHRED25 and the last 10 nt at the 5′-end of each read were removed to avoid quality drop-offs as well as the presence of low complexity tails, as suggested in the library kit manufacturer’s instructions. Reads were deduplicated using the dedupe tool implemented in the bbmap suite [28] and reads were assembled using MEGAHIT [29], applying the a set of kmers in the 21–119 range (21, k + 10, 119) and selecting a minimum contig length of 1500 nt. For long reads, the raw electric signals were base called using guppy v.6.2.1 in high confidence mode to generate fastq files, which were then merged per sample. Critical parameters, such as read length and average quality of the readouts, were evaluated using Nanoplot v1.40.2 [30]. Nanopore reads were screened for the presence of chimeric regions using yacrd v0.6.2 [31] and only reads passing coverage and chimeric filters were further used as input for read correction based on racon, using the ‘normal’ mode with a window size of 500 nt. Filtered nanopore reads were then assembled using metaFlye [32].

### 2.5. Identification and Analysis of Viral Sequences

To identify putative viruses we applied the Metaphage v.2 beta pipeline (https://github.com/MattiaPandolfoVR/MetaPhage/ (accessed on 1 May 2023)); [33], with several modifications. The databases utilised for the data analysis have been downloaded as indicated (all databases have been accessed on 1 May 2023): Inphared v.Jan2022 (https://github.com/RyanCook94/inphared), Phigaro v2.3.0 (https://github.com/bobeobibo/phigaro), KEGG (ftp://ftp.genome.jp/pub/db/kofam/archives/2019-03-20/), Pfam v.32 (https://pfam.xfam.org (ftp://ftp.ebi.ac.uk/pub/databases/Pfam/releases/Pfam32.0/)), VOG release 94 (http://vogdb.org/ (http://fileshare.csb.univie.ac.at/vog/vog94/)), Virsorter2 v2.2.3 (https://osf.io/v46sc/download), checkV v1.5 (https://portal.nersc.gov/CheckV/), CAT/BAT (tbb.bio.uu.nl/bastiaan/CAT_prepare/CAT_prepare_20210107.tar.gz) and NCBI nr (ftp://ftp.ncbi.nlm.nih.gov/blast/db/FASTA/nr.gz, accessed on 1 February 2023). PHROGs, CARD and VFDB databases were downloaded from https://zenodo.org/record/7563578/files/pharokka_v1.2.0_database.tar.gz. Five different tools were used to predict viral sequences, including virsorter2 v2.2.3 [34], deepvirfinder v.1.0 [35], phigaro [36], VIBRANT v1.2.1 [37] and VirFinder v1.1 [38], setting a minimal length of 2 kb for the tools allowing this parameter. Cd-hit-est [39] was used to dereplicate the resulting putative viral sequences applying 0.95 of similarity over 0.85 of contig length, thus providing viral Operational Taxonomic Units (vOTUs) as output. The vOTUs were further filtered with checkV v0.8.1 [40] to trim possible host genes and handle duplicate segments of circular contigs, removing the vOTUs with no detected viral genes. To evaluate the abundances of the vOTUs in the different samples, the quality-trimmed reads were mapped back on the vOTUs using bowtie v1.3.1 [41]. For Nanopore-derived vOTUs, minimap2 [42] was used for read back-mapping with standard parameters (-x ava-ont). The counts of mapped reads were used to compute relative abundance values as Count Per Millions (CPMs), thus normalizing for the contig length and the total mapped reads in each virome dataset. A rarefaction analysis using 6,895,930 reads per sample has been performed with the R package vegan v2.6. The taxonomic classification of the vOTUs was assessed by vpf-tools [43], which uses viral protein families for taxonomic determination and by CAT v5.0.4 [44], based on prodigal v2.6.3 [45] and DIAMOND v2.0.6 [46]. To evaluate the result of vpf-tools, we considered the taxonomic assignation with a confidence score (CS) of ≥0.2 and membership ratio (MR) of ≥0.2, as previously reported [42]. Functional annotation of the vOTUs was performed with pharokka v1.2.1 [47] based on the proteins predicted with prodigal, using PHROGs (38,880 protein orthologous groups), CARD (5170 Antimicrobial Resistance Detection Models) and VFDB (32,439 virulence factor-related genes), as reference databases applying an E-value to 0.01 to detect conserved domains. Predicted proteins were also blasted against the NCBI nr database (downloaded February 2023) using DIAMOND. The alpha diversity was computed across samples using a dedicated R script, as described in the Metaphage pipeline. Putative auxiliary metabolic genes were identified with VIBRANT. Data comparison and plotting were performed with R studio 23.03.0 based on R-4.2.1 using the following packages: tidyverse v2.0.0, magrittr v2.0.3, phyloseq v1.42.0, plotly v4.10.2 and matrixStats v1.0.0.

## 3. Results

### 3.1. Comparison between Unamplified and Randomly Amplified Benthic Abyssal Viromes

Shotgun short-read sequencing of four DNA libraries generated from pg to ng of virus-enriched unamplified (NA) or amplified (A) DNA samples yielded 2.678 billion reads, which were reduced to 1.327 billion reads by removing exact replicates of the same read (i.e., deduplication, Table 1). Non-amplified samples were characterized by a higher number of identical reads, resulting in a lower read number after deduplication (57.3 M for 39NA, 211.0 M for 57NA), compared to amplified samples (488.0 M for 39A and 471.0 M for 57A). Read assemblies resulted in 15,199, 79,174, 35,891 and 62,878 contigs longer than 1.5 kb for samples 39NA, 39A, 57NA, and 57A, respectively (Table 1). These contigs were separately used as input for five different bioinformatic tools allowing for virus identification based on machine learning or protein similarity approaches. Overall, a total of 88,086 unique contigs were identified as putative viruses, called vOTUs after dereplication. In more detail, 4 out of 5 tools identified more than 40,000 vOTUs, while phigaro identified a few hits only (Appendix A). Although 11,463 vOTUs were identified by all 4 tools, the majority of vOTUs were predicted by single tools or by different combinations of them. CheckV was used to further screen the putative vOTUs for the presence of provirus contamination, duplicated segments of circular sequences and to remove contigs with no viral gene detected, resulting in 24,828 high-confidence vOTUs, which defined the virome of the 2 investigated benthic abyssal sites. In all the analysed samples the sequencing depth was adequate to reach saturation (Appendix A).

Back-mapping of the reads on the 24,828 vOTUs was used to determine their relative abundances in the different samples, to evaluate the viral diversity among samples as well as the taxonomic composition of the viromes in terms of Baltimore classification, family, and genus. Notably, the number of the reads mapped on the vOTUs revealed that the amplified DNA samples were considerably enriched in viral sequences (38.5 and 49.9% of total reads for sample 39A and 57A, respectively), compared to the non-amplified samples (4.4 and 5.8%, for sample 39NA and 57NA, respectively, Table 2). Applying a cut off 3 CPMs to remove poorly represented vOTUs, the total number of vOTUs per sample was slightly higher in the non-amplified samples (11,031, 12,354, 10,346, and 10,393 counted vOTUs for 39NA, 57NA, 57A, and 39A samples, respectively; Table 2). Accordingly, viral richness values estimated by Chao1 as well as values of viral diversity by Shannon and Fisher indexes were higher in non-amplified than in amplified samples (Table 2).

### 3.2. Taxonomic Composition of the Benthic Abyssal Viromes

According to the taxonomic classification of vOTUs carried out with vpf-tools, we identified ssDNA and dsDNA viruses belonging to 22 viral families, including *Microviridae* (12,022), *Myoviridae* (352), *Siphoviridae* (324), *Podoviridae* (174), *Circoviridae* (28), *Parvoviridae* (3), *Poxviridae* (11), *Marseilleviridae* (3), *Phycodnaviridae* (18), and *Papillomaviridae* (3) as the 10 most represented families (Appendix A). Considering the abundances per sample, vOTUs with no Baltimore classification were the dominant group, followed by ssDNA viruses (Figure 1). dsDNA viruses were mainly detectable in the non-amplified samples (16 and 11% of the total abundance for sample 39NA and 57NA, respectively), whereas their relative abundance dropped below 2.2 and 0.97% for sample 39A and 57A, respectively. At the family level, the most abundant vOTUs in all samples could not be taxonomically classified (54–73%), followed in decreasing abundance by those identified as *Microviridae*, *Podoviridae*, *Myoviridae,* and *Siphoviridae* (Figure 2).

The heat map constructed on the most abundant vOTUs (>1000 CPMs) clustered the samples by DNA preparation method, nevertheless revealing hits with opposite trends among samples (Figure 3a). Notably, none of the relevant viral families (defined as the ones with average CPMs > 1000) resulted in being enriched by the amplification step, with the most represented family (*Microviridae*) maintaining the same relative abundance between the non-amplified and amplified samples. Conversely, the unclassified vOTUs increased their abundance in the amplified samples by 24% (Figure 3b,c).

The size distribution of the whole virome was characterized by two main peaks, corresponding to 2.2 and 4.4 kb (Figure 4a), which can be associated to uncharacterized vOTUs and *Microviridae*, respectively (Figure 4b).

Considering the genus-level taxonomic classification, 65–73% of the vOTUs remained unclassified, followed by *Clamydiamicrovirus* (18–27%) and T4-like phages (2–3%), with similar relative abundances in amplified and non-amplified samples (Figure 5 and Appendix A). Differently, Vp5-like, P22-like, Lambdavirus, and Bpp1-like vOTUs showed higher relative abundances in the non-amplified samples.

A total of 91 vOTUs, referring mostly to non-classified vOTUs (*N* = 71) and *Microviridae* (*N* = 14), displayed an average abundance > 1000 CPM in the 4 short-read datasets. Notably, 14 of these vOTUs were classified at genus level as putative *Clamydiamicrovirus*. Following a different taxonomic classification approach based on CAT/BAT, we noticed some annotation inconsistencies (Appendix A) as we identified 6 vOTUs similar to McMurdo Ice Shelf pond-associated circular DNA virus 8 and 16 vOTUs similar to *Circovirus* spp. Although not listed among the most abundant hits, we identified vOTUs likely referring to the family *Fuselloviridae*, as for instance vOTU_578 (length 27.8 kb, 66 predicted ORFs) which was erroneously classified under the “dsDNA/*Myoviridae*” category by the classification tool, although two ORFs showed similarity (<40% of aminoacidic similarity) to a *Nitrosopumilus* spindle-shaped virus. Finally, the functional annotation of the 151,800 proteins encoded by the 24,828 high-confidence vOTUs revealed a majority of “uncharacterized proteins” (96%), followed by tail (*N* = 2756), head and packaging (*N* = 2237) and DNA, RNA and nucleotide metabolism (*N*= 2572, Appendix A). A total of 135 proteins were identified as possible auxiliary metabolic genes (AMGs) by VIBRANT (Appendix A).

### 3.3. Output of the Long Read Sequecing of the Amplififed Virome

To further investigate the effect of the amplification process and evaluate the possibility of adopting single molecule sequencing to quickly profile a virome sample, we subjected the amplified DNA of site 57 to Nanopore sequencing. The resulting 301,522 reads (N50 = 9.6 kb, median length = 6.2 kb) were reduced to 36,152 error-corrected, non-chimeric reads and assembled into 1009 contigs (N50 = 9.9 kb). Applying the same pipeline as for short reads, a total of 555 vOTUs were identified, with the uncharacterized hits being the most abundant (46%), followed by vOTUs belonging to *Microviridae* (25%) and *Myoviridae* (21%) families. The size distribution of these vOTUs showed two-peaks, with the first peak (4.6 kb) reasonably associated to *Microviridae*, whereas the second peak was associated with both *Microviridae* and *Myoviridae* (Appendix A). In terms of taxonomic composition, the comparison of the long-read virome with the paired datasets obtained from short reads revealed a preserved ratio of ss/dsDNA viruses with the non-amplified dataset (Figure 1) and a similar distribution of families (Figure 2).

## 4. Discussion

Information on viral diversity in benthic abyssal ecosystems (>4000 m depth) and the contribution of ds- and ssDNA viruses is still extremely limited. In the present study, we analysed the viromes of two benthic abyssal sites of the North-eastern Atlantic Ocean using different sample preparation and sequencing approaches and evaluated their performance to investigate viral diversity, including the contribution of ssDNA and dsDNA viruses.

Starting from 10 ng of amplified DNA or from the unamplified DNA in the picogram-range, we successfully prepared sequencing libraries using the Accel-NGS 1S Plus kit. This kit has been previously tested with DNA quantities diluted up to picograms as well as from 100 ng of virome DNA [48], allowing the capture of both ss- and dsDNA molecules [49,50]. The higher number of PCR cycles required for the picogram DNA quantities resulted in a higher number of identical reads in the unamplified samples, compared to the amplified ones, which need to be removed to avoid possibly assembly and mapping biases [51].

By integrating different bioinformatic tools, we retrieved a total of 24,828 vOTUs, with the amplified samples considerably enriched in viral sequences (ca. 40–50% of total sequences in amplified samples vs. <6% in the non-amplified samples).

Overall, the taxonomic classification of vOTUs based on viral-specific profiles allowed the identification of 22 viral families, with *Microviridae*, *Podoviridae*, *Myoviridae,* and *Siphoviridae* being the most represented ones, as reported in other deep-sea sediment viromes [2,9,11,13]. The bimodal size distribution of our viromes likely depends upon the great abundance of *Microviridae* that we recovered, plus taxonomically uncharacterized small viruses contributing to defining the smaller size peak. Irrespective of the procedure used to prepare the viromes, most of vOTUs had no matches in the database, reinforcing previous findings that benthic deep-sea ecosystems represent a hotspot of novel viruses [2,8,52]. Despite the lower number of viral reads in the non-amplified samples, viral diversity in terms of represented vOTUs and detected viral families was higher than in the amplified samples. The higher percentage of dsDNA-vOTUs in the unamplified samples (11–16%) compared to the amplified samples (ca. 1–2%) represents a further consistent difference. It is known that DNA amplification step by Phi29 polymerase can determine a preferential amplification of circular ssDNA to dsDNA [53], and thus, the relative abundance of ssDNA viral sequences can be overrepresented [2,8]. On the other side, studies indicate that dsDNA viruses can be an important component of the benthic deep-sea viral diversity. For example, a metagenomic study performed on deep-sea sediments of the Southwest Indian Ocean without prior separation of viral particles highlighted almost exclusively the presence of dsDNA viruses mostly affiliated with the families *Myoviridae*, *Podoviridae* and *Siphoviridae* [11]. Other findings obtained from the analyses of viromes generated with different procedures from other benthic deep-sea ecosystems revealed the presence of a relevant fraction of ssDNA viruses and dsDNA viruses [2,8,9]. This discrepancy could be due to the natural variability of the investigated deep-sea ecosystems, but also to the applied methodologies, which could profoundly influence the assessment of viral diversity. In our study, the virome was mainly composed of ssDNA viruses irrespective of the amplification step. Since the adopted methodology for library preparation allows for the recovery of ssDNA molecules, this might have contributed to a “more genuine” assessment of the different components. Intriguingly, we also obtained a similar ratio of ds/ssDNA viruses by applying long-read sequencing to one amplified DNA sample. This is possibly linked to limitations in the assembly process of amplified datasets, which failed to correctly resolve complex contigs, suggesting that long-read sequencing can be used to profile the virome composition before massive sequencing. Advancing the technology towards low-input DNA samples will likely make long-read sequencing pivotal for metagenomic/viromic analyses [21].

In our study, the genus-level taxonomic annotation by vpf-tools reveals that the majority of vOTUs in all samples were unclassified, mirroring the result obtained at the family level. Notably, the genus *Chlamydiamicrovirus* belonging to the *Microviridae* family was the most abundant both in amplified and non-amplified samples. Its presence is not surprising since its natural host (i.e., *Chlamydiae)* has already been reported in different environments, including deep-sea sediments from the Arctic Mid-Ocean Ridge [54]. Other identified viral genera showed differential abundances, typically with higher values in the non-amplified samples, as reported at the family-level classification. Although genus-level classification has been obtained using previously utilised confidence scores [43], these findings should be viewed with caution since the limited length of the viral contigs may affect the correct taxonomic classification [55].

Among viruses known to infect prokaryotic hosts, we found an almost complete genome of a putative archaeal virus belonging to the family *Fuselloviridae*, exclusively present in unamplified samples. Virus-induced lysis of archaea has been reported to be a key process in deep-sea sediments, as it contributes to the decrease of up to one-third of the total prokaryotic biomass [56]. Since only a few archaeal viruses have been studied in detail so far [57], shedding light on the viruses that contribute to archaea dynamics and related ecosystem processes is critical to advancing science.

In the two investigated abyssal sites, a number of different contigs were identified as circoviruses. Although not abundant, their presence, which has been already reported in different benthic deep-sea habitats [58,59], poses questions regarding their ecological relevance as viruses infecting metazoans [60]. Among other possible eukaryotic viruses, we identified 18 vOTUs belonging to the *Phycodnaviridae* family, which includes NCLDV infecting photosynthetic eukaryotes (proposed order, “Megavirales”). These viruses have been previously reported to occur in high abundance in deep-sea ecosystems as result of their transport associated with sinking particles from the surface waters [61]. A few vOTUs, likely representing fragments of viruses belonging to *Iridoviridae*, *Herpesviridae,* and *Marseilleviridae* families were also identified.

Mirroring the presence of abundant unclassified vOTUs, also the large majority of proteins encoded by the vOTUs were uncharacterized. Only a minimal part of these proteins putatively harboured auxiliary metabolic functions, mainly related to carbohydrate and aminoacid metabolism. Nowadays, a “functional” role of viromes has been proposed [62,63]. Although the most represented AMGs that we found recalled the ones reported in a deep sea virome of the Mariana trench [64], the limited extension of both our dataset and of the retrieved AMGs prevents us from drawing general conclusions regarding their functional role.

## 5. Conclusions

Overall, our findings indicate that the analysis of viromes obtained from abyssal sediments by using direct sequencing (without amplification step) can provide a broad overview of viral richness and the composition of the viral assemblages in terms of ssDNA- and dsDNA-vOTUs. At the same time, the contextual use of a random amplification step of the same samples and Nanopore long-read technology can improve the assessment of viral assemblages and, thus, it represents a suitable approach for expanding our knowledge on viral diversity in the largest ecosystem on Earth and towards a better understanding of their interactions with potential hosts, both prokaryotes and eukaryotes.

## Figures and Tables

**Figure 1 viruses-15-02282-f001:**
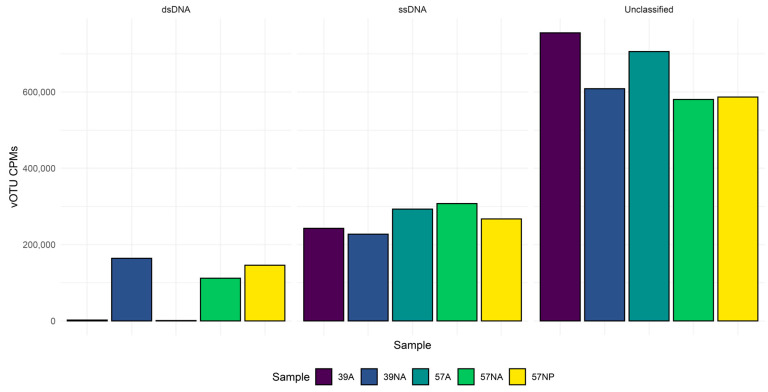
Taxonomic classification of the viromes in the two investigated benthic abyssal sites. The abundance of the vOTUs corresponding to dsDNA, ssDNA, and uncharacterized Baltimore groups are reported for the five reconstructed viromes. NA: non-amplified; A: amplified; NP: Nanopore.

**Figure 2 viruses-15-02282-f002:**
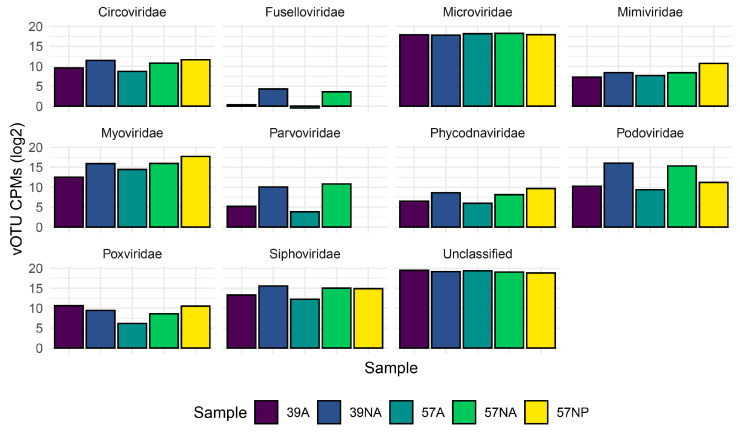
Taxonomic classification of the viromes in the two investigated benthic abyssal sites at the family level. The abundance of the vOTUs corresponding to the most represented families is reported in a log2 scale for the five reconstructed viromes. NA: non-amplified; A: amplified; NP: Nanopore.

**Figure 3 viruses-15-02282-f003:**
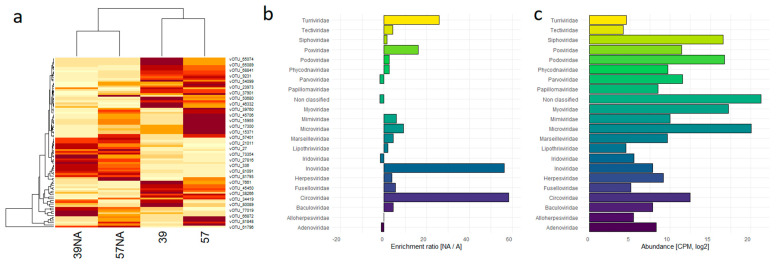
Taxonomic distribution of vOTUs found in the five samples. (**a**) Heat map based on the 91 vOTUs with CPMs > 1000 and depicted by their relative abundances in the 4 samples. Dark-red boxes referred to the most abundant vOTUs; (**b**) enrichment ratio, computed by viral family, between non-amplified (NA) and amplified (A) samples; (**c**) total abundances of the identified viral families are reported as CPM in a log scale. The color codes of the families are maintained between panel (**b**,**c**).

**Figure 4 viruses-15-02282-f004:**
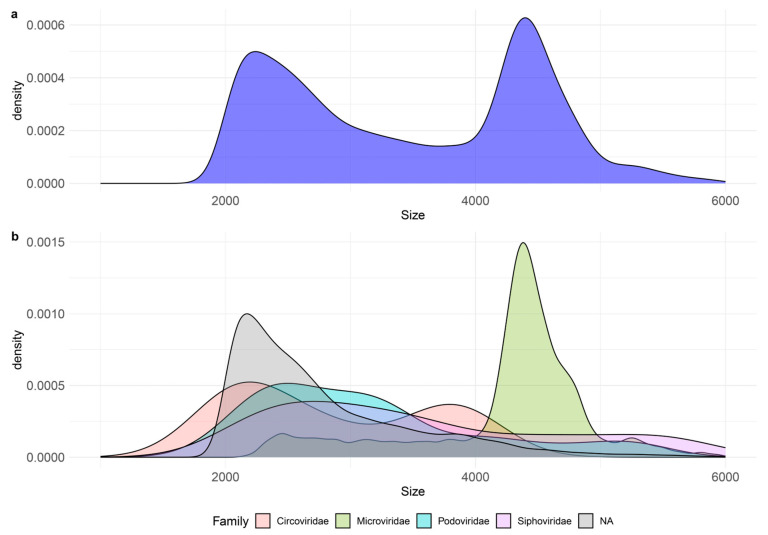
Size distribution of the benthic abyssal viromes reconstructed with short reads sequencing. (**a**) Total size distribution and (**b**) size distribution of the vOTUs belonging to the five most represented families.

**Figure 5 viruses-15-02282-f005:**
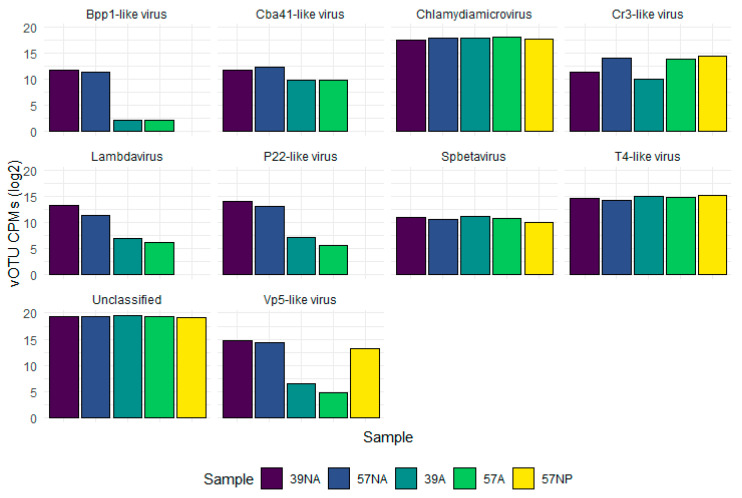
Taxonomic classification at the genus level of the viromes in the two investigated abyssal sites. The abundance of the vOTUs corresponding to the 10 most represented genera is reported for the 5 reconstructed viromes in a log2 scale. NA: non-amplified; A: amplified; NP: Nanopore.

**Table 1 viruses-15-02282-t001:** Summary of samples, sequencing and assembly details. Sample ID and description, sequencing technology, the starting amount of DNA used to prepare the libraries, number of deduplicated reads, number of contigs longer than 1.5 kb and size of the longest contig (kb) are reported for each sample.

Sample ID	Description	Technology	Starting DNA Quantity	No. of Deduplicated Reads [×10^6^]	No. of Contigs Longer Than 1.5 kb	Longest Contig (kb)
39NA	Site 39, not amplified	Illumina	20–50 pg	157.3	15,199	72,980
39A	Site 39, amplified	Illumina	10 ng	488.0	79,174	151,451
57NA	Site 57, not amplified	Illumina	20–50 pg	211.0	35,891	135,810
57A	Site 57, amplified	Illumina	10 ng	471.0	62,878	135,810
		Nanopore	1 µg	0.331	1009	

**Table 2 viruses-15-02282-t002:** Overview of the benthic abyssal viromes. The number of the high-confidence vOTUs identified, together with the percentages of mapped reads, viral richness estimated by Chao1 and alpha diversity values, calculated according to Shannon and Fisher indexes, are reported per sample. NA: non-amplified; A: amplified.

Sample ID	No. of vOTUs	Total Reads Mapped Back on vOTUs	Richness Chao1	Alpha Diversity Shannon	Alpha Diversity Fisher
39NA	11,031	4.4%	10,626	7.11	1661.12
39A	10,393	38.5%	9662	6.96	1487.58
57NA	12,354	5.8%	11,760	7.30	1875.05
57A	10,346	49.9%	9659	6.86	1486.90

## Data Availability

The data generated for this study are available in the NCBI SRA database under the project accession ID PRJNA909474.

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
