# Peer review of "Viral Diversity in Benthic Abyssal Ecosystems: Ecological and Methodological Considerations"

_viruses, 2023, doi:10.3390/v15122282_

Round 1
Reviewer 1 Report
Comments and Suggestions for Authors
The manuscript discusses how different methodological approaches can significantly impact the assessment of viral diversity in sediments. The authors employed state-of-the-art methods and effectively compared their results with recent literature. Their results are straightforward and provide valuable insights into the viral ecology within deep sediments.
However, I have a few general comments and some specific suggestions. Firstly, it is unclear why this paper is submitted to the 'Animal Viruses' section, as most of the discovered viral genomes belong to bacteriophages. Secondly, it appears that the number of sample replicates used to validate the methods is not adequate or not mentioned in M&M section. Additionally, I suggest shifting the focus from the importance of viruses in the water column in the opening paragraph to those present in sediment.
Here are some specific comments:
Introduction
- Line 58: This is the first point at which the reader becomes aware that the study is specifically focused on sediments, rather than the water column. It would be beneficial to provide more specific and precise information to clarify that the study primarily concerns testing methods in sediment, as 'benthos' has a broader meaning.
Materials and Methods
- Line 104: Have you conducted sample testing in triplicates? Since you are evaluating the method, a certain number of repetitions is necessary.
Discussion
- In general, while discussing the ecology of the identified viruses is essential, it's also crucial to emphasize the impact of the methods used in the discovery of viruses in sediments. Your conclusion neatly summarizes the primary objectives of the study. Consider dividing the conclusion into three parts and aligning the discussion accordingly.
- Lines 406-414: The text in this section appears unnecessary since your study did not investigate the viral cycle affecting archaea. It's important to briefly mention it and highlight its significance but I wouldn’t take it further.
Author Response
We thank the two anonymous reviewers for providing comments on our manuscript. We have addressed all issues raised by the two Reviewers, as outlined below and we are now submitting the revised version of the paper for your consideration.
Reviewer#1. The manuscript discusses how different methodological approaches can significantly impact the assessment of viral diversity in sediments. The authors employed state-of-the-art methods and effectively compared their results with recAent literature. Their results are straightforward and provide valuable insights into the viral ecology within deep sediments. However, I have a few general comments and some specific suggestions. Firstly, it is unclear why this paper is submitted to the 'Animal Viruses' section, as most of the discovered viral genomes belong to bacteriophages. Secondly, it appears that the number of sample replicates used to validate the methods is not adequate or not mentioned in M&M section. Additionally, I suggest shifting the focus from the importance of viruses in the water column in the opening paragraph to those present in sediment.
We wish to thank the Reviewer for appreciating our effort in providing new insights into the viral diversity of deep-sea sediments by using comparative methodological approaches.
The Reviewer is right that most of the identified viral genomes in our manuscript belong to phages and that the submission of our manuscript under the “animal viruses” section is misleading. Therefore, we kindly ask the Editor to modify the section to which our manuscript has been submitted by shifting to the “bacterial viruses” section. Regarding the number of replicates, we collected sediment samples from two benthic deep-sea sites at ca. 1.8 km from each other and at the same depth (ca. 4,700), thus these samples represent actual replicates of the deep-sea area investigated (i.e., the Abyssal Hill). Before virus extraction, we pooled a similar amount of sediment collected by three independent multi-corer deployments at each site to recover a representative amount of viral DNA, which can provide a more realistic view of the actual viral diversity present therein. In the amended version we better explained such issues. Aiming to obtain the most informative results from the two sample pairs analysed in this study, we increased the sequencing effort and produced a total of 2.678 billion reads and, eventually, 24,828 high-confidence vOTUs. Of course, the availability of more replicates could further support our conclusions, but unfortunately sample availability was limited, as typically occurs when benthic abyssal systems (at >4,000 m depth) are investigated. According to the Reviewer suggestion we modified the introduction by omitting the general aspects dealing with viruses in the water column and directly focusing on viruses present in deep-sea sediments.
- Line 58: This is the first point at which the reader becomes aware that the study is specifically focused on sediments, rather than the water column. It would be beneficial to provide more specific and precise information to clarify that the study primarily concerns testing methods in sediment, as 'benthos' has a broader meaning.
Thanks for the suggestion. We modified the introduction in order to immediately introduce the issues related to viruses present in deep-sea sediments.
- Line 104: Have you conducted sample testing in triplicates? Since you are evaluating the method, a certain number of repetitions is necessary.
We sequenced two set of paired samples representative of each of study sites
- In general, while discussing the ecology of the identified viruses is essential, it's also crucial to emphasize the impact of the methods used in the discovery of viruses in sediments. Your conclusion neatly summarizes the primary objectives of the study. Consider dividing the conclusion into three parts and aligning the discussion accordingly.
We think the conclusions we provided already contain the main ecological and methodological outputs which can be used for further comparative studies dealing with viral diversity in deep-sea sediments.
- Lines 406-414: The text in this section appears unnecessary since your study did not investigate the viral cycle affecting archaea. It's important to briefly mention it and highlight its significance but I wouldn’t take it further
Thanks, we have reduced this part.
Reviewer 2 Report
Comments and Suggestions for Authors
The present manuscript analyzed and evaluated viral diversity of samples obtained from benthic abyssal sites using different treatment and methods. It could be considered for publication after revisions.
Suggestions for Revision.
1. Clarifying what your mean, in order to avoid confusion. Such as:
1) line 83, “the five DNA viromes” was not clear in Introduction and the following parts (e.g. The column "Sample ID" in Table 1 contains only four DNA samples).
2) line 161, the title of 2.4 indicated database download, but the content was not described in this section.
3) Some statements were not be sure. E.g., 24,828 vOTUs were obtained and considered as high-confidence vOTUs. However, it is not known which data the following analysis was based. Only in the Microviridae “identified viruses were 12,022, nearly half of the high-confidence vOTUs” (line239⁓259). Why the authors stated that “the majority of the vOTUs were unknown”?
4) line 422, “and literature therein)”?
2. Due to the recent advances about viruses in aquatic ecosystems are too brief in Introduction, it is need to cite relevant references (e.g. “Recent insights into aquatic viruses: Emerging and reemerging pathogens, molecular features, biological effects, and novel investigative approaches, Water Biology and Security, 2022, 1(4), 100062”).
3. The “Results” section (line219⁓337) should be divided into different sub-sections with sub-titles to make the section more clarified.
4. Standards were needed for the layout. For figure, the figure legends should be located below each figure.
5. Because there are various data and analysis in the results, listing a concise table that includes sample, treatment methods, and the different analyzing results is required to guide readers through your research paper
Author Response
We thank the two anonymous reviewers for providing comments on our manuscript. We have addressed all issues raised by the two Reviewers, as outlined below and we are now submitting the revised version of the paper for your consideration.
Reviewer#2. The present manuscript analyzed and evaluated viral diversity of samples obtained from benthic abyssal sites using different treatment and methods. It could be considered for publication after revisions.
We thank the Reviewer for the positive comments on our manuscript and for providing suggestions for improving the quality of our work.
Clarifying what your mean, in order to avoid confusion. Such as:
line 83, “the five DNA viromes” was not clear in Introduction and the following parts (e.g. The column "Sample ID" in Table 1 contains only four DNA samples).
We clarified this point in the text. The five viromes were referred to the number of sequencing readouts that we generated (4 Illumina and 1 Nanopore), but now we removed it for sake of clarity.
line 161, the title of 2.4 indicated database download, but the content was not described in this section.
We described the used databases in section 2.5. Accordingly, we have modified the headings.
Some statements were not be sure. E.g., 24,828 vOTUs were obtained and considered as high-confidence vOTUs. However, it is not known which data the following analysis was based. Only in the Microviridae “identified viruses were 12,022, nearly half of the high-confidence vOTUs” (line239⁓259). Why the authors stated that “the majority of the vOTUs were unknown”?
Thanks for this comment. As correctly pointed out by the Reviewer, around half of the vOTUs were taxonomically classified. However, if we consider the abundances of the vOTUs, most of the viral abundances in all samples referred to unknown vOTUs. We amended the text to improve readability (see line 267-270).
line 422, “and literature therein)”?
Removed from the amended version of the manuscript.
Due to the recent advances about viruses in aquatic ecosystems are too brief in Introduction, it is need to cite relevant references (e.g. “Recent insights into aquatic viruses: Emerging and reemerging pathogens, molecular features, biological effects, and novel investigative approaches, Water Biology and Security, 2022, 1(4), 100062”).
Thanks for the suggestion. Accordingly, we included the citation suggested by the Reviewer and for accomplishing also the Reviewer 1 requests we focused primarily on viruses present in deep-sea sediments.
The “Results” section (line219⁓337) should be divided into different sub-sections with sub-titles to make the section more clarified.
Ok, thanks.
Standards were needed for the layout. For figure, the figure legends should be located below each figure.
Thanks for the suggestion which has been accomplished in the amended version of the manuscript.
Because there are various data and analysis in the results, listing a concise table that includes sample, treatment methods, and the different analyzing results is required to guide readers through your research paper
Thanks for the suggestion which has been taken into account by the details contained in the Table 1 which can help the reader to understand what has been done on sediment samples by using the different methodologies.
Round 2
Reviewer 1 Report
Comments and Suggestions for Authors
Dear authors,
I have no further comments. All the issues within my scope of expertise have been adequately addressed. I find the paper to be publishable.
Best regards